# A Modulation Layer to Increase Neural Network Robustness Against Data Quality Issues

## Abstract

Data quality is a common problem in machine learning, especially in high-stakes settings such as healthcare. Missing data affects accuracy, calibration, and feature attribution in complex patterns. Developers often train models on carefully curated datasets to minimize missing data bias; however, this reduces the usability of such models in production environments, such as real-time healthcare records. Making machine learning models robust to missing data is therefore crucial for practical applications. While some classifiers naturally handle missing data, others, such as deep neural networks, are not designed for unknown values. We propose a novel neural network modification to mitigate the impacts of missing data. The approach is inspired by neuromodulation that is performed by biological neural networks. Our proposal replaces the fixed weights of a fully-connected layer with a function of an additional input (reliability score) at each input, mimicking the ability of cortex to up- and down-weight inputs based on the presence of other data. The modulation function is jointly learned with the main task using a multi-layer perceptron. We tested our modulating fully connected layer on multiple classification, regression, and imputation problems, and it either improved performance or generated comparable performance to conventional neural network architectures concatenating reliability to the inputs. Models with modulating layers were more robust against degradation of data quality by introducing additional missingness at evaluation time. These results suggest that explicitly accounting for reduced information quality with a modulating fully connected layer can enable the deployment of artificial intelligence systems in real-time settings.

## 1 Introduction

Despite the enormous academic and industrial interest in artificial intelligence, there is a large gap between model performance in laboratory settings and real-world deployments. Reports estimate that over 75% of data science and artificial intelligence projects do not make it into production (VentureBeat, 2019; Sagar, 2021; Chen and Asch, 2017). One difficult transition from the laboratory is handling noisy and missing data. Errors in predictor data and labels (Northcutt et al., 2021) at the training stage are well understood to produce poor pattern recognition with any strategy; garbage-in garbage-out. In the statistical learning literature, the effects of inaccurate and missing data on simple classifiers such as logistic regression is particularly well understood (Ameisen, 2020). As a result, datasets intended to train high-accuracy models are often carefully curated and reviewed for validity (Ameisen, 2020; Xiao et al., 2018). However, when faced with noisy data from a new source, these models may fail (L'Heureux et al., 2017). One special case is convolutional neural networks for machine vision; augmenting the dataset with partially obscured inputs has been shown to increase the network's ability to match low-level patterns and increases accuracy (Zhong et al., 2020). No similar results with masking have been shown in tabular data, to our knowledge.

These challenges are even more pronounced in applications that require high reliability and feature pervasive missing data at inference time, such as healthcare (Chen and Asch, 2017; Xiao et al., 2018). Electronic health records (EHR) can contain a high percentage of missing data both at random (keyboard entry errors, temporarily absent data due to incomplete charting) and informative or missing-not-at-random (MNAR) data (selective use of lab tests or invasive monitors based on

observed or unobserved patient characteristics). Medical measurements also have non-uniform noise; for instance, invasive blood pressure measurement is more accurate than non-invasive blood pressure (Kallioinen et al., 2017). Another example is the medical equipment by different manufacturers that have various margins of error which affects the accuracy and hence the reliability of the measurement (Patel et al., 2007).

Mammalian brains have a distinct strategy to integrate multi-modal data to generate a model of the surrounding environment. They modify the impact of each input based on the presence and reliability of other signals. This effect can be observed dynamically in response to temporary changes in available inputs (Shine et al., 2019), as well as long-term as a compensation mechanism for permanent changes such as neural injuries (Hylin et al., 2017). For example, a human brain gives less weight to visual input in a dark environment and relies on prior knowledge and other sensory cues more. Unlike simply down-weighting low-accuracy data, replacement data with related information is up-weighted. This is usually modelled as a Bayesian inference process (Cao et al., 2019; Ernst and Bülthoff, 2004; Alais and Burr, 2004; Heeger, 2017). This modulation of different input is also observed in other organisms where the neural behavior of a neuron or a group of neurons can be altered using neuromodulators (Harris-Warrick and Marder, 1991). We used the inspiration from this process to design a fully-connected neural network layer with variable weights. Those weights could be modulated based on a variety of inputs, but we focus on input reliability as a modulating signal. This allowed us to train the neural network using datasets that are loosely preprocessed with a high incidence of missing data while achieving high performance. At inference time, the network is capable of producing accurate outputs despite signal degradation. A restricted structure of modulating inputs and effects on the modulated layer reduces the likelihood of severe over-fitting and complexity of the estimation problem.

## 2 RELATED WORK

The most obvious use case we propose for this structure is handling missing data. There is a vast literature on imputation, which also attempts to use alternative inputs to replace missing data. Classical simple methods of imputation include constant values (e.g. mean imputation), hot deck, k-nearest neighbor, and others (Buck, 1960). Single or multiple imputation using chained equations (Gibbs sampling of missing data) is popular due to its relative accuracy and ability to account for imputation uncertainty (Azur et al., 2011). More advanced yet classic methods have seen relative success such as Bayesian ridge regression (MacKay, 1992) and random forest imputation (Stekhoven and Bühlmann, 2012). Deep learning-based imputation has been used recently using generative networks (Beaulieu-Jones and Moore, 2017; McCoy et al., 2018; Lu et al., 2020; Lall and Robinson, 2021; McCoy et al., 2018; Mattei and Frellsen, 2019; Yoon et al., 2018; Ivanov et al., 2019) and graph networks (You et al., 2020). Our modulation approach can be incorporated into autoencoder architectures to improve their performance and stability in data imputation, but it also provides the flexibility of skipping the imputation step altogether when the task performed does not require imputation (i.e. classification) thus skipping a preprocessing step and saving processing time.

Incorporating uncertainty measurements into deep neural networks has also been approached with Bayesian deep learning methods, (Wang and Yeung, 2016; Wilson, 2020) which has a complex, assumption laden structure using probabilistic graphical models. One simpler variation of Bayesian deep learning is the Gaussian process deep neural network which assigns an uncertainty level at the output based on the missing data so that inputs with greater missingness lead to higher uncertainty (Bradshaw et al., 2017). Our method makes use of meaningful missingness patterns as opposed to treating it as a problem that leads to lower confidence in outputs. Our approach is superficially similar to attention mechanisms: the lower quality inputs receive less importance, but attention networks employ a complex feedback mechanism to assign the attention distribution using the input sequence and the query and is thus difficult to scale for long time-varying inputs (Kim et al., 2017).

## 3 METHODS

### 3.1 ARCHITECTURE

A fully connected layer has a transfer function of

$$h_{out} = f(\mathbf{W} \cdot h_{in} + b), \tag{1}$$

where $h_{in}$ is the input to the layer, $\mathbf{W}$ is the weight matrix , $\mathbf{b}$ the bias and $f$ the non-linearity function. $\mathbf{W}$ is optimized during training and fixed at inference. We propose a modulated fully connected layer (MFCL) where weights are made variable by replacing $\mathbf{W}$ by $\mathbf{W_{mod}}$ (Figure 1) where

$$\mathbf{W_{mod}} = g(m), \tag{2}$$

where $m$ is the modulating signal input and $g$ is the function that is defined by a multilayer perceptron. Another variant of the MFCL (MFCL+) adds a skip connection to the transfer step and modifies the weights ($\mathbf{W_0}$) of a starting network,

$$\mathbf{W_{mod+}} = g(m) + \mathbf{W_0}. \tag{3}$$

The latter architecture aims at being more adaptive to datasets with less or no missing values and could avoid potential instabilities by having the entire layer weights variable.

### 3.2 EXPERIMENTS

We assessed the performance of the MFCL and MFCL+ layers in classification, regression, and imputation tasks. These experiments used modulating signals of missing value flags and input reliability values of noisy data. We can think of missing values as a special case included in reliability where *missing* implies completely unreliable measurement. For the sake of clarity, we test the cases of missing values and noisy values separately rather than combining them. For baseline comparison, we employed models with matching architectures while swapping the first fully-connected layer with a MFCL/MFCL+. Base architectures were guided by previous best performing models in the literature. Modulation network architectures were optimized using a grid search. A complete description of the architectures is elaborated in the appendix.

### 3.3 DATASETS

The motivating dataset for our experiments derives from operating room data from Barnes Jewish Hospital's Anesthesiology Control Tower project (ACTFAST) spanning 2012–2018. The [Name redacted for anonymity] IRB approved this study and granted a waiver of informed consent. The dataset contains preoperative measurements of medical conditions, demographics, vital signs, and lab values of patients as well as postoperative outcomes that were used as labels for supervised learning including 30-day mortality, acute kidney injury, and heart attack. The ACTFAST dataset was used in previous studies for prediction of 30-day mortality (Fritz et al., 2019), acute kidney injury, and

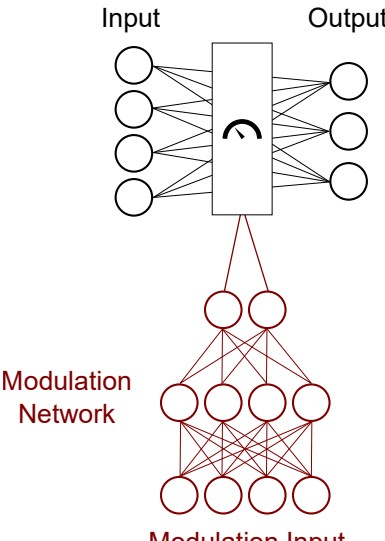

Figure 1: Schematic of modulated fully connected layer. The weights of the fully connected layers are modulated by the output of the modulation network.

other complications (Cui et al., 2019; Abraham et al., 2021; Fritz et al., 2018). For predictors, we utilized a subset of the input features of preoperative vital signs and lab values (15 variables). Table 1 (appendix) shows a list of variables used and the missing percentages. Table 2 (appendix) shows the distribution of outcome values, which have a large imbalance between positive and negative samples. We also used the Wisconsin Breast Cancer dataset for classification of tumors from features extracted from a fine needle aspirate of breast mass image (Mangasarian et al., 1995). We used the Boston housing prices dataset (Harrison Jr and Rubinfeld, 1978) as a regression example. Each of the above datasets was also used for imputation tasks.

### 3.4 CLASSIFICATION TASK

We ran five experiments for classification using the ACTFAST and Breast Cancer datasets. Four experiments utilized the missing flags as a modulating signal; in the last experiment, we utilized input reliability as a modulating signal. Reliability was quantified by the standard deviation of the noise that was artificially added to the signal. We tested the MFCL and MFCL+ in the place of fully-connected (FC) layers at the input level.

#### 3.4.1 BASELINES

The baseline classifiers for the ACTFAST and Breast Cancer datasets were four MLPs with matching hidden layer structures that were fed imputed values using different algorithms, namely: chained regression with Bayesian ridge regularization (Scikit Learn Iterative Imputer), missForest algorithm (Stekhoven and Bühlmann, 2012), GAIN (Yoon et al., 2018), and VAEAC (Ivanov et al., 2019). We also tested one graphical network model GRAPE (You et al., 2020). The last model applied constant value imputation (mean value) in addition to concatenating an indicator variable for missing values at the input layer (FC+Mod).

**ACTFAST** We built three classifiers to predict 30-day Mortality, Acute Kidney Injury (AKI), and Heart Attack from the preoperative input features (Table 1). We used the datasets with the inherent missing data for training and then tested the trained models with additional missingness artificially introduced in both random and non-random fashions. Non-random missingness was introduced by removing the largest values and by all datapoints of features (discussed below).

**Breast Cancer** For the classifier with missing flags as modulating signal, we introduced non-random missingness into the training dataset by removing the highest quartile of each variable. At the testing phase we evaluated each model with additional missingness similar to the ACTFAST classifiers.

For the classifier with reliability signal, we utilized the complete dataset but added Gaussian noise with zero mean and variable standard deviation (SD) where the SD values were sampled from a uniform distribution between 1 and 10 standard deviations of each variable. The higher end of SD values is very large, simulating a spectrum of noisy to essentiall missing data. Breast cancer dataset includes values that were measured from images of a fine needle aspitate of a breast mass to describe cell nuclei characteristics. Extracted data includes mean values as well as standard errors and worst value measurements. Due to this nature of variables and them including error rate values, we selected 10 variables representing the mean value measurements only for this experiment. We tested using a 20% test split on the same noisy data.

### 3.5 IMPUTATION TASK

We ran three experiments for imputation by an auto-encoder using the ACTFAST, Breast Cancer, and Boston datasets. For the ACTFAST experiment, we utilized the predictor features described above. We added the MFCL and MFCL+ layers in the place of FC layers at the inputs of an autoencoder imputation system. All parameters of training were similar to the baseline autoencoder described below.

#### 3.5.1 BASELINES

The baseline autoencoder for imputation was trained by adding artificial missingess to the input values at random at a ratio of 25%. The loss function at the output layer calculated the mean squared error between the output values and the original values of the artificially removed values. The naturally missing data was included in the training dataset but not included in the loss function due to the absence of a known value to compare to. Then the weights were optimized using an Adam optimizer with learning rate 0.01 and a learning rate scheduler that reduced the learning rate after five epochs of loss not improving. We ran 30 epochs of training for the ACTFAST models with a batch size of 64, and 200 epochs for the Breast Cancer and Boston models. The models were tested using variable artificial missingness.

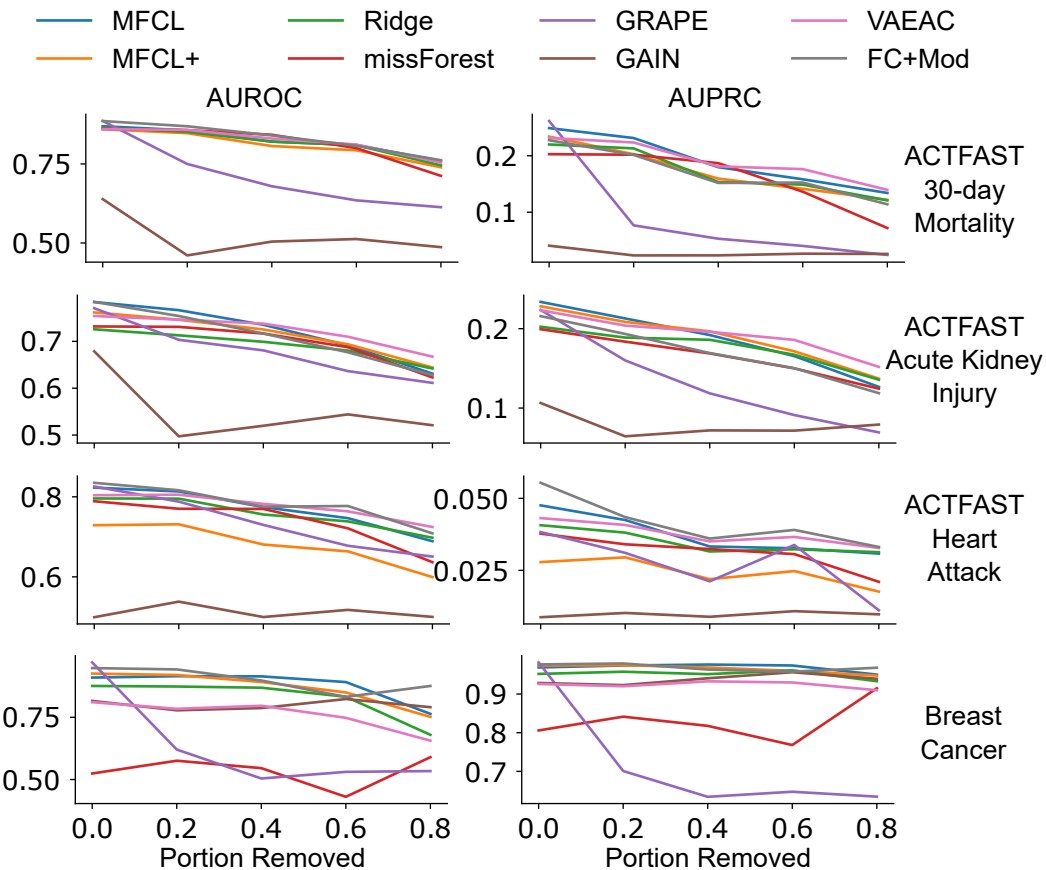

Figure 2: Performance on classification tasks with artificial introduction of random missingness to increasing portions of the input data (Error bars represent 95% confidence intervals).

## 3.6 REGRESSION TASK

We ran one experiment for regression using the Boston dataset where, similar to previous experiments, we added the MFCL and MFCL+ layers in the place of FC layers at the inputs of the networks and used missing flags as modulating signal.

### 3.6.1 BASELINES

For the regressor with missing flags as modulating signal, we introduced artificial non-random missingness into the dataset for training the regressor by removing the highest quartile of each variable. At the testing phase we evaluated each model with additional missingness similar to the classification tasks. The baseline regression networks were two MLPs with matching hidden layer structures. The first one had input variables imputed in a preprocessing step using chained regression (Iterative Imputer). The second one applied constant value imputation (mean value) in addition to concatenating the missing value (input reliability) at the input layer (FC+Mod).

## 3.7 PERFORMANCE EVALUATION

We performed an 80:20 training test split for each dataset to measure the performance for each of the architectures. We performed all our additional missingness tests only on the test split of the datasets. For classification tasks, we utilized area under receiver operating curve (AUROC) and area under precision and recall curve (AUPRC). In the training phase, binary cross-entropy loss was utilized as a cost function. For regression and imputation tasks, we utilized mean squared error loss value as

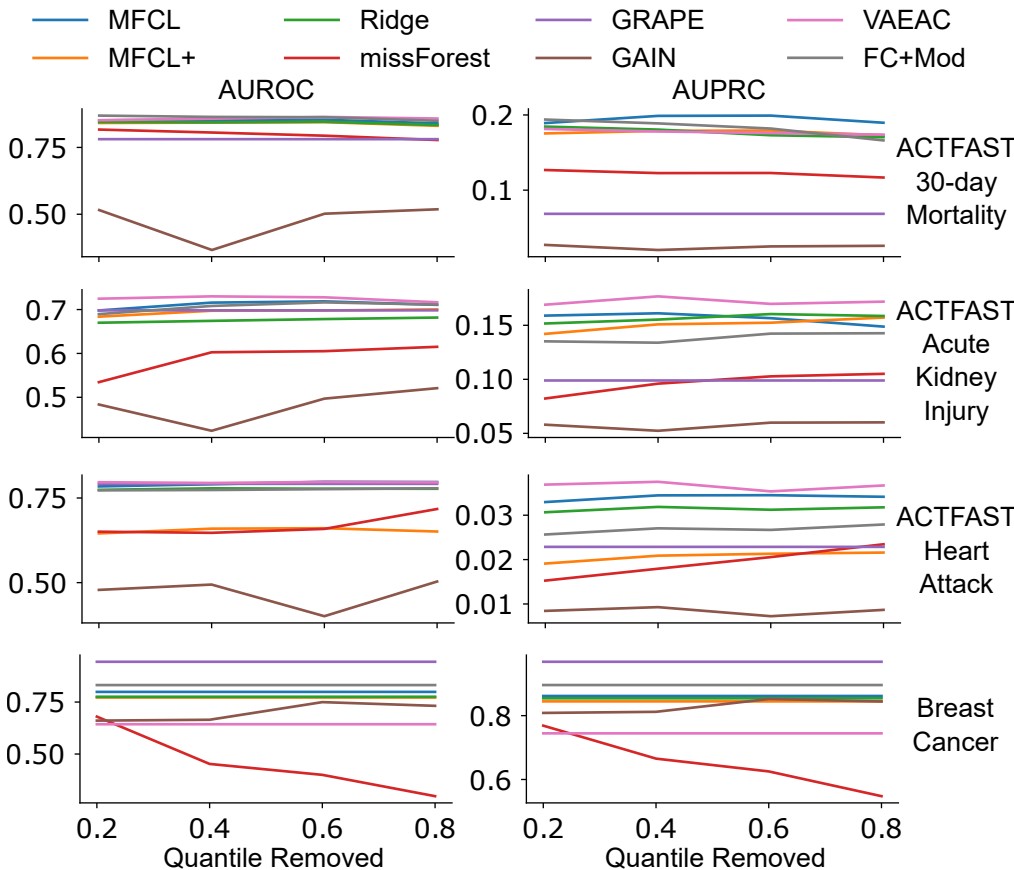

Figure 3: Performance on classification tasks with artificial introduction of non-random missingness (achieved by removing input values above the specified quantile for each feature) (Error bars represent 95% confidence intervals).

both the training cost function and the test performance evaluation metric.

To compute the margins of error, we conducted 1000 folds of paired bootstrapping for each experiment and computed the 95% confidence intervals for each test case. We show the complete results with confidence intervals in the supplementary information. To test for statistical significance, we calculated repeated measure ANOVA on the bootstrapping results followed by paired t-test between different model pairs with correction for false discovery rate using Benjamini/Yekutieli method.

## 4  RESULTS

### 4.1  CLASSIFICATION WITH MISSING VALUES

Figure 2 plots the test performance of baseline and modified classifiers as a function of additional random missingness. FC+Mod architecture provided the best performance along with VAEAC (on few conditions only) with MFCL and MFCL+ coming close to both of them with a very comparable performance (small effect sizes).

Figure 3 plots classifier performance under increasing amounts of non-random missingness by removing values higher than 80%, 60%, 40%, and 20% quantile from each variable. In this testing case, the performance gain by MFCL is more pronounced where MFCL is outperforms other architectures significantly on the AUPRC of mortality and comes at a close second for Acute Kidney

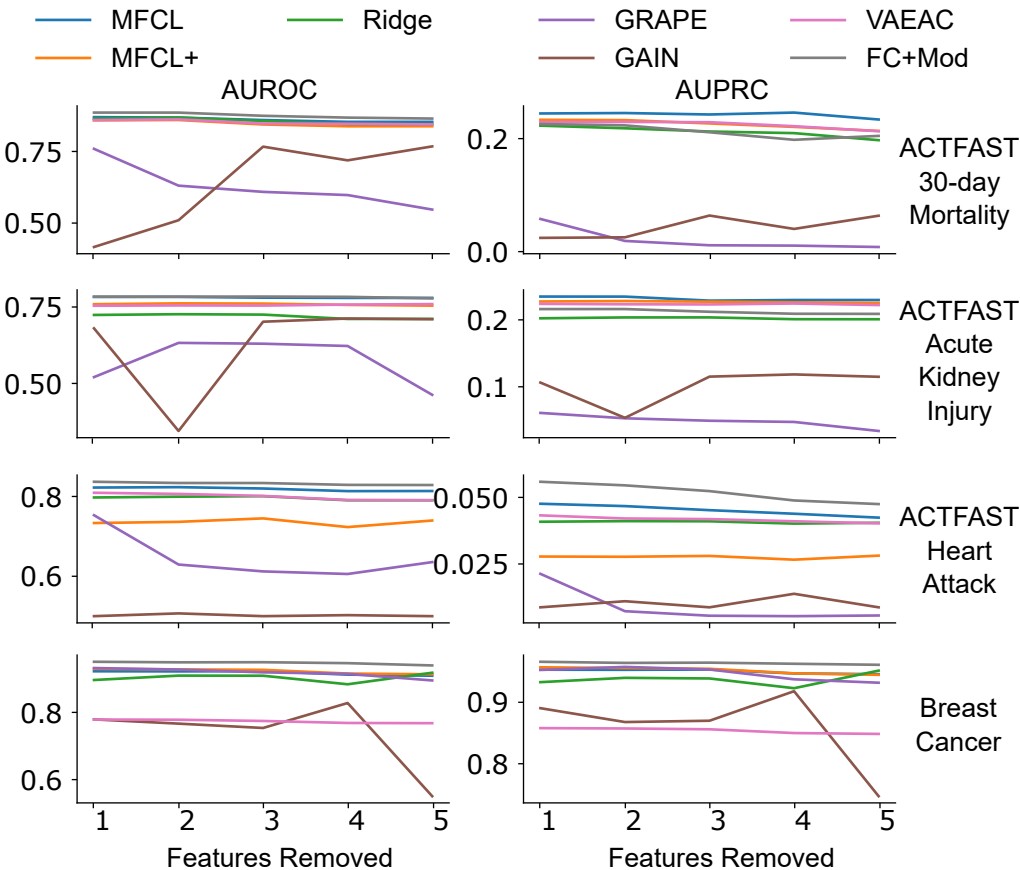

Figure 4: Performance on classification tasks with artificial introduction of non-random missingness (achieved by removing complete features from the test dataset) (Error bars represent 95% confidence intervals).

Injury to VAEAC. DNN+Mod outperforms all networks on the Breast Cancer dataset. This could be due to the relatively small size of the dataset leading to greater of overfitting on larger networks.

Figure 4 shows another example of non-random missingness by removing complete features. This can happen in reality when attempting to test a model using a dataset coming from a different source where some of the measurements are not conducted due to cost or other reasons. We tested this on all the methods except for the missForest which failed when a feature value were completely absent. Here MCFL shows the best performance for AUPRC for Mortality and AKI.

We can see that the modulating architectures are the most consistent especially on the precision and recall measures which is a powerful representative of performance in highly imbalanced data. While some models outperform in certain conditions, they are not as reliable as they fail in other conditions. To our surprise the DNN+Mod model also performed quite well which is a simpler variant of a modulation architecture however it appeared to favor AUROC performance which is not ideal for highly imbalanced datasets.

These results show that MFCL and MFCL+ networks give additional robustness against large quantitites of non-random missingness while still performing strongly well where missingness is low, especially in precision which is most important in highly imbalanced datasets such as ACTFAST.

## 4.2 AUTOENCODER IMPUTATION WITH MISSING VALUES

We tested imputation on three different datasets (Figure 4). We tested the imputation networks by introducing 10% missingness in the test datasets and measuring the mean squared error. We found that

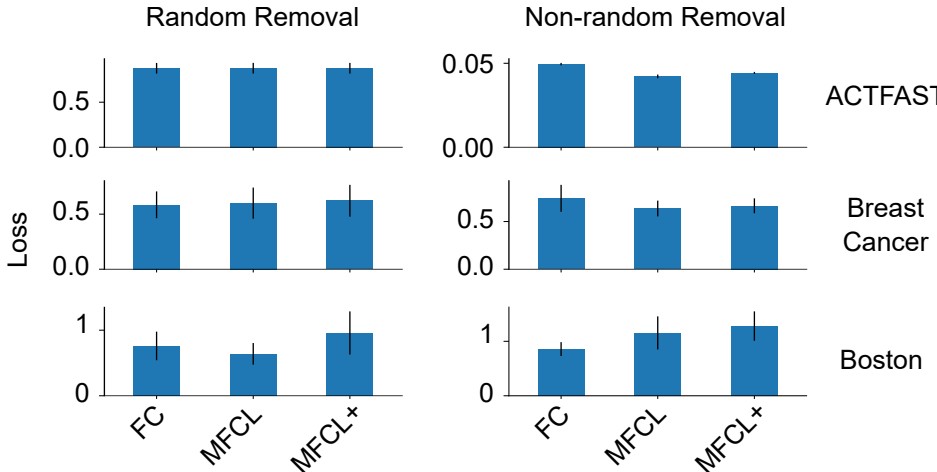

Figure 5: Performance on imputation tasks with artificial introduction of missing data in random (left column) and non-random (right column) fashions (Error bars represent 95% confidence intervals).

for ACTFAST, the addition of modulation layer did not add much to the imputation performance in comparison to the normal autoencoder for random removal. However, for the non-random removal, the MFCL/MFCL+ layers showed significantly lower loss indicating higher performance. We also note that the range of losses in this dataset is very different between random and non-random removal which indicates the usefulness of meaningful missingness patterns in predicting the latent representation of the data. It appears that all the networks were able to learn that represnetation but with our architectures learning is better. For other datasets, we found no significant differences which could possibly due to the small size of datasets increasing the overfitting possibility in comparison with the ACTFAST dataset which is orders of magnitude larger.

### 4.3 REGRESSION WITH MISSING VALUES

We utilized the Boston housing dataset as an example of a regression problem with missing data (Figure 6 Supplementary information). In this task, while it appears that MFCL outperformed all other networks significantly in all conditions with VAEAC coming as a close second. Interestingly, the performance either remained the same or improved with more data removed. This could possibly be due to uncorrelated inputs that have very little shared information especially given the high absolute values of loss.

### 4.4 CLASSIFICATION WITH INPUT VALUES WITH VARIABLE RELIABILITY

We tested the modulation layer where input reliability is used as a modulating signal instead of missing flags (Figure 7 Supplementary information). In this condition, the MFCL+ outperformed all other architectures over both AUROC and AUPRC measures significantly. This task performance could be better assessed with real data as it becomes available.

### 5 DISCUSSION AND CONCLUSION

We propose a new layer for artificial neural networks inspired by biological neuromodulation mechanisms (Harris-Warrick and Marder, 1991). It allows the neural network to alter its weights and thus behavior based on the modulating signal (figure 8 supplementary materials). Our experiments showed that, when added to standard architectures, modulating input layers make predictions more robust to missing and low quality data. In classification, regression, and imputation tasks modulation was most useful when non-random missingness was introduced. However, there was not a consistent benefit to the MFCL versus MFCL+ layer. This could be due to the large number of parameters especially in the last layer of the modulating network which scales with the product

of the number of inputs and outputs of the fully connected layer.

Our testing procedure was limited by multiple factors discussed below. First, due to the novelty and flexibility of this model, there are many possible combinations for hyperparameters to explore. In order to limit the hyperparameter search space, we fixed the main network architecture and only varied the modulation network hyperparameters, but in practice there may be interactions between the hyperparameters of the two component networks. One other limitation is the lack of availability of large open tabular datasets with high missingness which limits the ability to generalize our findings. To make our experiments with informative missingness comparable across features, we restricted our input space to numeric variables and discarded categorical variables. Although our method could be applied to missing categorical variables, usually creating a "missing" level is fairly effective. Small technical modifications would also be required to modulate all features derived from encoding a categorical variable in the same way. We expect that future experiments with new real-world datasets will better characterize the performance of this method.

We tested the application of the modulation process only in fully connected layers which are limited by nature in the types of data that they can handle. We intend to test the inclusion of modulation into other architectures such as convolutional layers and gated-recurrent units. It is important to address the issue of the high number of parameters in the modulation network. We did not search over regularization strategies of the modulation network, which may further improve its performance. This is a main benefit of the modulation strategy compared to the conventional strategy of concatenating a missing data indicator to the inputs, which doubles the input space and complicates the search for appropriate architectures and regularization strategies. By separating the two architectures, we can learn a plug-and-play modification for any classification task on the same inputs. It might also prove beneficial to integrate the modulation layer into the existing deep learning imputation models which is an aspect of future investigation especially with its relative stability across different tasks in comparison to the other methods.

One extension of our approach is to add the MFCL in locations in the network beyond the input layer. Preliminary experiments placing MFCL layers deep in the autoencoder experiments did not yield visible improvement (data not shown). The modulating signal could also be any input signal such as context signal in a context switching task which could yield this network useful in multi-task reinforcement learning problems among many other applications (Jovanovich and Phillips, 2018). It can also be useful in compressing multi-task networks by compressing the multiple outputs into one with modulating input acting as a switch to change behavior of the network based on the task in question (Kendall et al., 2018; Chen et al., 2018; Li et al., 2020).

In conclusion, we have demonstrated that a modulation architecture could benefit in training neural networks in avenues where data quality is an issue. It can lead to advance the field of MLOps which is concerned with the integration of machine learning systems into production environments and thus fulfilling a big portion of the potential of artificial intelligence systems in advancing state-of-the-art technologies.

## 6    ETHICS STATEMENT

It is important, to understand the possible dangers that lie behind the unethical use of such an architecture where it cause amplification of certain societal biases that are visible in the data. In lower resource settings, marginalized groups have been observed to have more missing data (Chen et al., 2020). Prediction methods not accounting for missing data can produce inaccurate results for these groups and hence, disadvantaging them. Therefore, methods that explicitly account for missing data instead of discarding the data are better in terms of social equity. On the other hand, non-transparency of neural networks, especially that use only small amount of data points for feature values can lead to feature-wise bias amplification (Leino et al., 2018). A solution to mitigate these issues would be to perform a contextual post-processing check on the prediction results. Overall, we believe the proposed algorithm's positive societal impacts outweigh the negative ones.

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

## A APPENDIX

The models were all built and tested using pytorch 1.6 (https://pytorch.org/) and run on a GeForce GTX 1080 GPU (Nvidia Corportation, United States). The base networks were designed based on knowledge of previous literature that utilized the datasets we used in this paper. For the MFCL and MFCL+ layers architectures, we tested a small subset of modulation architectures on the ACTFAST 30-day mortality data. We then fixed the architecture for other ACTFAST tasks. We decreased the size of the architecture on other datasets to avoid overfitting. To perform the hyperparameter search, we split the data into training, validation, and test sets using a 70:10:20 ratio. Tested architectures for the modulation layer are as follows:

- 1 hidden layer with 8 neurons
- 2 hidden layers with 8 neurons each
- 3 hidden layers with 8 neurons each
- 3 hidden layers with 8-4-8 neurons
- 4 hidden layers with 8 neurons each
- 5 hidden layers with 8 neurons each

We found little differences (none significant) and selected the highest performing architecture. We then combined the training and validation sets to generate a new training set that was used on the final model training. We tried to avoid a large number of hyperparameter tuning as we attempt to test the stability of the new architecture in less than optimal conditions.

The architectures and training parameter of the networks presented are as follows

**Classification tasks**   ACTFAST
The base network architecture were as follows:

- Number of hidden layers: 2
- Hidden layers' architecture: 8-4
- 3 hidden layers with 8 neurons each
- Hidden layers activation function: ReLU
- Output layer activation function: Sigmoid
- Dropout: After first hidden layer with rate 0.5
- Modulation network architecture: 3 hidden layers with 8 neurons each
- Modulation layer location: Hidden layer 1
- Modulation network dropout: After first hidden layer with rate 0.5

**Classification tasks**   Breast Cancer (including both missing data and noised data tasks)
The base network architecture were as follows:

- Number of hidden layers: 2
- Hidden layers' architecture: 4-2
- 2 hidden layers with 8-4 neurons
- Hidden layers activation function: ReLU
- Output layer activation function: Sigmoid
- Dropout: After first hidden layer with rate 0.5
- Modulation network architecture: 3 hidden layers with 8 neurons each
- Modulation layer location: Hidden layer 1
- Modulation network dropout: After first hidden layer with rate 0.0

**30-day Mortality task**  The training parameters were as follows

- Batch size: 64
- Number of epochs: 30
- Optimizer: Stochastic gradient descent
- Learning rate: 0.001
- Momentum: 0.9
- Learning rate scheduler: Multiply learning rate by a a factor of 0.25 when AUROC does not increase for five epochs.

**Acute kidney injury task**  The training parameters were as follows

- Batch size: 64
- Number of epochs: 50
- Optimizer: Stochastic gradient descent
- Learning rate: 0.001
- Momentum: 0.9
- Learning rate scheduler: Multiply learning rate by a a factor of 0.25 when AUROC does not increase for five epochs.

**Heart attack task**  The training parameters were as follows

- Batch size: 64
- Number of epochs: 80
- Optimizer: Stochastic gradient descent
- Learning rate: 0.001
- Momentum: 0.9
- Learning rate scheduler: Multiply learning rate by a a factor of 0.25 when AUROC does not increase for five epochs.

**Breast cancer task**  The training parameters were as follows

- Batch size: 64
- Number of epochs: 100
- Optimizer: Stochastic gradient descent
- Learning rate: 0.03
- Momentum: 0.9
- Learning rate scheduler: Multiply learning rate by a a factor of 0.25 when AUROC does not increase for five epochs.

**Imputation Tasks**  ACTFAST & Breast Cancer
The base network architecture were as follows:

- Number of hidden layers: 3
- Hidden layers' architecture: 10-5-10
- 2 hidden layers with 8-4 neurons
- Hidden layers activation function: ReLU
- Output layer activation function: Linear
- Dropout: After first hidden layer with rate 0.2
- Modulation network architecture: 3 hidden layers with 8 neurons each
- Modulation layer location: Hidden layer 1
- Modulation network dropout: After first hidden layer with rate 0.2

**Imputation Tasks**   Boston housing dataset
The base network architecture were as follows:

- Number of hidden layers: 3
- Hidden layers' architecture: 5-2-5
- 2 hidden layers with 8-4 neurons
- Hidden layers activation function: ReLU
- Output layer activation function: Linear
- Dropout: After first hidden layer with rate 0.2
- Modulation network architecture: 3 hidden layers with 8-4-8 neurons
- Modulation layer location: Hidden layer 1
- Modulation network dropout: After first hidden layer with rate 0.2

**Imputation Tasks**   Training parameters were as follows:

- Batch size: 64
- Number of epochs: ACTFAST: 30, Breast Cancer & Boston: 100
- Optimizer: Adam
- Learning rate: 0.01
- Betas: 0.9 and 0.999
- Learning rate scheduler: Multiply learning rate by a a factor of 0.25 when loss does not drop for five epochs.

**Regression Task**   Network architecture and training parameters were as follows:

- Number of hidden layers: 2
- Hidden layers' architecture: 4-2
- 2 hidden layers with 8-4 neurons
- Hidden layers activation function: ReLU
- Output layer activation function: Linear
- Dropout: After first hidden layer with rate 0.5
- Modulation network architecture: 2 hidden layers with 8-4 neurons
- Modulation layer location: Hidden layer 1
- Modulation network dropout: After first hidden layer with rate 0.5
- Batch size: 64
- Number of epochs: 100
- Optimizer: Stochastic gradient descent
- Learning rate: 0.001
- Momentum: 0.9
- Learning rate scheduler: Multiply learning rate by a a factor of 0.25 when loss does not drop for five epochs.

Table 1: Input variables and missing percentages in ACTFAST datasets.

| Input Variable | Missing Percentage | | |
|---|---|---|---|
| | 30-day Mortality N=67961 | AKI N=106870 | Heart Attack N=111888 |
| Systolic Blood Pressure | 58.5% | 57.3% | 57.5% |
| Diastolic Blood Pressure | 59.0% | 57.8% | 58.1% |
| Heart Rate | 1.2% | 1.3% | 1.3% |
| $SpO_2$ | 1.0% | 1.1% | 1.1% |
| Alanine Transaminase | 67.5% | 65.4% | 66.1% |
| Albumin | 67.3% | 65.1% | 65.8% |
| Alkaline Phosphatase | 67.5% | 65.4% | 66.1% |
| Creatinine | 22.4% | 23.8% | 26.1% |
| Glucose | 20.2% | 21.7% | 23.4% |
| Hematocrit | 20.4% | 22.4% | 24.1% |
| Partial Thromboplastin Time | 61.5% | 59.3% | 60.2% |
| Potassium | 22.0% | 23.3% | 25.0% |
| Sodium | 21.9% | 23.3% | 25.0% |
| Urea Nitrogen | 22.0% | 23.4% | 25.1% |
| White Blood Cells | 22.2% | 23.9% | 26.2% |

Table 2: Output variables imbalance in classfication datasets.

| Output Variable | Positive Percentage |
|---|---|
| 30-day Mortality | 2.3% |
| Acute Kidney Injury | 6.1% |
| Heart Attack | 0.9% |
| Breast Cancer | 62.7% |

Table 3: AUROC mean values and confidence intervals for the 30-day Mortality classification task for random removal followed by non-random removal and feature removal conditions.

| | MFCL | MFCL+ | Ridge | missForest | GRAPE | GAIN | VAEAC | FC+Mod |
|---|---|---|---|---|---|---|---|---|
| 0.0 | 0.87 (0.04) | 0.86 (0.04) | 0.86 (0.04) | 0.86 (0.04) | 0.89 (0.04) | 0.31 (0.05) | 0.86 (0.04) | 0.89 (0.03) |
| 0.2 | 0.86 (0.04) | 0.85 (0.04) | 0.85 (0.04) | 0.86 (0.04) | 0.75 (0.06) | 0.50 (0.05) | 0.86 (0.04) | 0.87 (0.03) |
| 0.4 | 0.82 (0.04) | 0.81 (0.04) | 0.82 (0.04) | 0.84 (0.04) | 0.68 (0.06) | 0.53 (0.06) | 0.83 (0.04) | 0.84 (0.04) |
| 0.6 | 0.81 (0.05) | 0.79 (0.05) | 0.81 (0.04) | 0.80 (0.05) | 0.63 (0.07) | 0.47 (0.05) | 0.81 (0.05) | 0.81 (0.04) |
| 0.8 | 0.76 (0.05) | 0.74 (0.05) | 0.75 (0.05) | 0.71 (0.05) | 0.61 (0.08) | 0.50 (0.02) | 0.75 (0.06) | 0.76 (0.05) |
| 0.8 | 0.85 (0.04) | 0.84 (0.04) | 0.85 (0.04) | 0.82 (0.04) | 0.78 (0.05) | 0.52 (0.02) | 0.85 (0.04) | 0.87 (0.04) |
| 0.6 | 0.85 (0.04) | 0.84 (0.04) | 0.84 (0.04) | 0.81 (0.04) | 0.78 (0.05) | 0.37 (0.06) | 0.86 (0.04) | 0.86 (0.04) |
| 0.4 | 0.85 (0.04) | 0.84 (0.04) | 0.85 (0.04) | 0.79 (0.05) | 0.78 (0.05) | 0.50 (0.03) | 0.86 (0.04) | 0.86 (0.04) |
| 0.2 | 0.84 (0.04) | 0.83 (0.04) | 0.83 (0.04) | 0.78 (0.05) | 0.78 (0.05) | 0.52 (0.04) | 0.86 (0.04) | 0.85 (0.04) |
| 1 | 0.87 (0.04) | 0.86 (0.04) | 0.86 (0.04) | NaN | 0.76 (0.04) | 0.44 (0.02) | 0.86 (0.04) | 0.89 (0.03) |
| 2 | 0.87 (0.04) | 0.86 (0.04) | 0.87 (0.04) | NaN | 0.63 (0.09) | 0.39 (0.03) | 0.86 (0.04) | 0.89 (0.03) |
| 3 | 0.86 (0.04) | 0.84 (0.04) | 0.86 (0.04) | NaN | 0.61 (0.10) | 0.76 (0.04) | 0.85 (0.04) | 0.87 (0.03) |
| 4 | 0.85 (0.04) | 0.84 (0.05) | 0.85 (0.04) | NaN | 0.60 (0.10) | 0.51 (0.03) | 0.85 (0.04) | 0.87 (0.04) |
| 5 | 0.85 (0.04) | 0.84 (0.05) | 0.85 (0.04) | NaN | 0.55 (0.09) | 0.79 (0.04) | 0.84 (0.04) | 0.86 (0.04) |

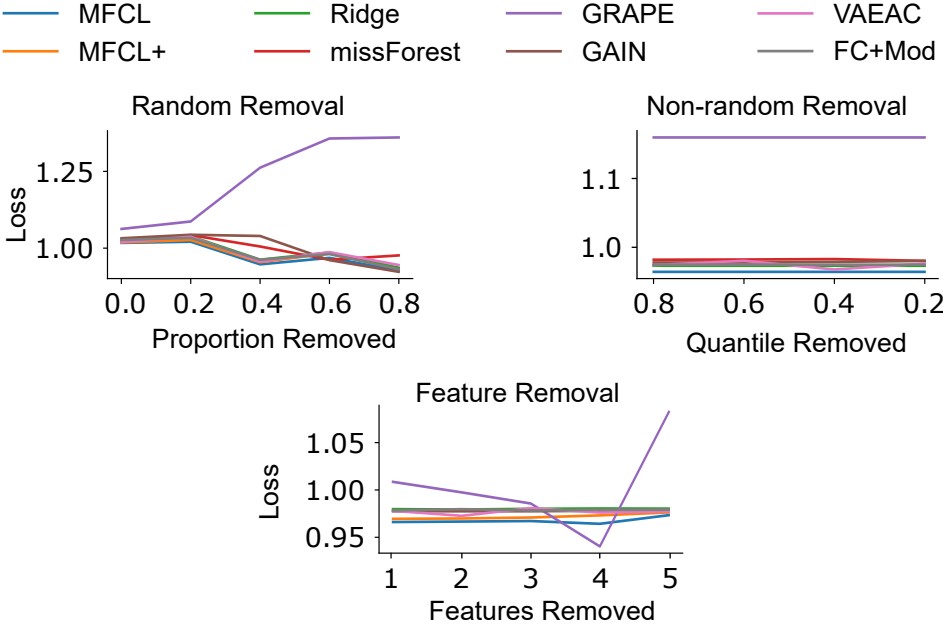

Figure 6: Performance on regression task in the Boston housing prices dataset with artificial introduction of missing data in random (left column) and non-random (right column) fashions (Error bars represent 95% confidence intervals).

Table 4: AUPRC mean values and confidence intervals for the 30-day Mortality classification task for random removal followed by non-random removal and feature removal conditions.

|      | MFCL | MFCL+ | Ridge | missForest | GRAPE | GAIN | VAEAC | FC+Mod |
|------|------|-------|-------|-----------|-------|------|-------|--------|
| 0.00 | 0.25 (0.08) | 0.23 (0.08) | 0.22 (0.07) | 0.20 (0.07) | 0.26 (0.09) | 0.02 (0.00 ) | 0.23 (0.07) | 0.23 (0.06) |
| 0.20 | 0.23 (0.07) | 0.20 (0.07) | 0.21 (0.07) | 0.20 (0.07) | 0.08 (0.03) | 0.02 (0.01) | 0.22 (0.08) | 0.20 (0.06) |
| 0.40 | 0.18 (0.07) | 0.16 (0.06) | 0.15 (0.06) | 0.19 (0.06) | 0.05 (0.03) | 0.03 (0.01) | 0.18 (0.07) | 0.15 (0.05) |
| 0.60 | 0.16 (0.06) | 0.14 (0.05) | 0.15 (0.06) | 0.14 (0.06) | 0.04 (0.03) | 0.02 0.00 ) | 0.18 (0.07) | 0.15 (0.06) |
| 0.80 | 0.13 (0.06) | 0.12 (0.05) | 0.12 (0.05) | 0.07 (0.03) | 0.02 (0.01) | 0.03 (0.01) | 0.14 (0.06) | 0.11 (0.04) |
| 0.80 | 0.19 (0.07) | 0.18 (0.06) | 0.18 (0.07) | 0.13 (0.05) | 0.07 (0.02) | 0.03 (0.01) | 0.18 (0.07) | 0.19 (0.06) |
| 0.60 | 0.20 (0.06) | 0.18 (0.06) | 0.18 (0.06) | 0.12 (0.05) | 0.07 (0.02) | 0.02 (0.01) | 0.18 (0.06) | 0.19 (0.06) |
| 0.40 | 0.20 (0.07) | 0.18 (0.06) | 0.17 (0.06) | 0.12 (0.05) | 0.07 (0.02) | 0.03 (0.01) | 0.18 (0.06) | 0.18 (0.06) |
| 0.20 | 0.19 (0.06) | 0.17 (0.06) | 0.17 (0.06) | 0.12 (0.05) | 0.07 (0.02) | 0.03 (0.01) | 0.17 (0.06) | 0.17 (0.05) |
| 1 | 0.24 (0.08) | 0.23 (0.08) | 0.22 (0.07) | NaN | 0.06 (0.02) | 0.02 (0.01) | 0.23 (0.07) | 0.23 (0.06) |
| 2 | 0.25 (0.08) | 0.23 (0.08) | 0.22 (0.08) | NaN | 0.02 (0.02) | 0.02 0.00 ) | 0.23 (0.08) | 0.22 (0.06) |
| 3 | 0.24 (0.08) | 0.23 (0.07) | 0.21 (0.07) | NaN | 0.01 (0.01) | 0.06 (0.02) | 0.23 (0.08) | 0.21 (0.06) |
| 4 | 0.25 (0.08) | 0.22 (0.07) | 0.21 (0.07) | NaN | 0.01 (0.01) | 0.02 0.00 ) | 0.22 (0.08) | 0.20 (0.06) |
| 5 | 0.23 (0.07) | 0.21 (0.07) | 0.20 (0.07) | NaN | 0.01 0.00 ) | 0.07 (0.02) | 0.21 (0.07) | 0.20 (0.06) |

Table 5: AUROC mean values and confidence intervals for the Acute Kidney Injury classification task for random removal followed by non-random removal and feature removal conditions

|  | MFCL | MFCL+ | Ridge | missForest | GRAPE | GAIN | VAEAC | FC+Mod |
|---|---|---|---|---|---|---|---|---|
| 0.00 | 0.78 (0.02) | 0.76 (0.03) | 0.73 (0.03) | 0.73 (0.03) | 0.77 (0.02) | 0.68 (0.02) | 0.75 (0.02) | 0.78 (0.02) |
| 0.20 | 0.77 (0.02) | 0.75 (0.02) | 0.71 (0.03) | 0.73 (0.03) | 0.70 (0.03) | 0.50 (0.01) | 0.75 (0.02) | 0.75 (0.02) |
| 0.40 | 0.74 (0.02) | 0.73 (0.03) | 0.70 (0.03) | 0.72 (0.03) | 0.68 (0.03) | 0.52 (0.03) | 0.74 (0.02) | 0.72 (0.02) |
| 0.60 | 0.69 (0.03) | 0.69 (0.03) | 0.68 (0.03) | 0.69 (0.03) | 0.64 (0.03) | 0.54 (0.02) | 0.71 (0.03) | 0.68 (0.03) |
| 0.80 | 0.63 (0.03) | 0.64 (0.03) | 0.64 (0.03) | 0.62 (0.03) | 0.61 (0.04) | 0.52 (0.03) | 0.67 (0.03) | 0.63 (0.03) |
| 0.80 | 0.71 (0.02) | 0.70 (0.03) | 0.68 (0.03) | 0.62 (0.03) | 0.70 (0.02) | 0.40 (0.03) | 0.72 (0.03) | 0.71 (0.02) |
| 0.60 | 0.72 (0.02) | 0.70 (0.03) | 0.68 (0.03) | 0.61 (0.03) | 0.70 (0.02) | 0.51 (0.02) | 0.73 (0.02) | 0.72 (0.02) |
| 0.40 | 0.72 (0.02) | 0.70 (0.03) | 0.67 (0.03) | 0.60 (0.03) | 0.70 (0.02) | 0.55 (0.02) | 0.73 (0.02) | 0.71 (0.02) |
| 0.20 | 0.70 (0.03) | 0.68 (0.03) | 0.67 (0.03) | 0.53 (0.03) | 0.70 (0.02) | 0.45 (0.02) | 0.72 (0.02) | 0.69 (0.03) |
| 1 | 0.78 (0.02) | 0.76 (0.02) | 0.72 (0.03) | NaN | 0.52 (0.03) | 0.68 (0.02) | 0.75 (0.02) | 0.78 (0.020) |
| 2 | 0.78 (0.02) | 0.76 (0.02) | 0.73 (0.03) | NaN | 0.63 (0.03) | 0.34 (0.03) | 0.76 (0.02) | 0.78 (0.020) |
| 3 | 0.78 (0.02) | 0.76 (0.02) | 0.73 (0.03) | NaN | 0.63 (0.03) | 0.70 (0.03) | 0.75 (0.02) | 0.78 (0.020) |
| 4 | 0.78 (0.02) | 0.76 (0.02) | 0.71 (0.03) | NaN | 0.62 (0.03) | 0.71 (0.02) | 0.76 (0.02) | 0.78 (0.02) |
| 5 | 0.78 (0.02) | 0.75 (0.03) | 0.71 (0.03) | NaN | 0.46 (0.03) | 0.71 (0.02) | 0.76 (0.02) | 0.78 (0.02) |

Table 6: AUPRC mean values and confidence intervals for the Acute Kidney Injury classification task for random removal followed by non-random removal and feature removal conditions.

|  | MFCL | MFCL+ | Ridge | missForest | GRAPE | GAIN | VAEAC | FC+Mod |
|---|---|---|---|---|---|---|---|---|
| 0.00 | 0.23 (0.04) | 0.23 (0.04) | 0.20 (0.03) | 0.20 (0.03) | 0.22 (0.04) | 0.11 (0.02) | 0.22 (0.04) | 0.22 (0.03) |
| 0.20 | 0.21 (0.03) | 0.21 (0.03) | 0.19 (0.03) | 0.18 (0.03) | 0.16 (0.03) | 0.06 (0.01) | 0.20 (0.03) | 0.19 (0.03) |
| 0.40 | 0.19 (0.03) | 0.20 (0.04) | 0.19 (0.03) | 0.17 (0.03) | 0.12 (0.03) | 0.07 (0.01) | 0.20 (0.03) | 0.17 (0.03) |
| 0.60 | 0.17 (0.03) | 0.17 (0.03) | 0.17 (0.03) | 0.15 (0.03) | 0.09 (0.02) | 0.07 (0.01) | 0.19 (0.03) | 0.15 (0.03) |
| 0.80 | 0.13 (0.02) | 0.14 (0.03) | 0.14 (0.03) | 0.12 (0.02) | 0.07 (0.02) | 0.08 (0.01) | 0.15 (0.03) | 0.12 (0.02) |
| 0.80 | 0.15 (0.03) | 0.16 (0.03) | 0.16 (0.03) | 0.11 (0.02) | 0.10 (0.01) | 0.06 (0.01) | 0.17 (0.03) | 0.14 (0.02) |
| 0.60 | 0.16 (0.03) | 0.15 (0.03) | 0.16 (0.03) | 0.10 (0.02) | 0.10 (0.01) | 0.06 (0.01) | 0.17 (0.03) | 0.14 (0.02) |
| 0.40 | 0.16 (0.03) | 0.15 (0.03) | 0.16 (0.03) | 0.10 (0.02) | 0.10 (0.01) | 0.05 (0.01) | 0.18 (0.03) | 0.13 (0.02) |
| 0.20 | 0.16 (0.03) | 0.14 (0.03) | 0.15 (0.03) | 0.08 (0.02) | 0.10 (0.01) | 0.06 (0.01) | 0.17 (0.03) | 0.14 (0.02) |
| 1 | 0.23 (0.04) | 0.23 (0.04) | 0.20 (0.03) | NaN | 0.06 (0.01) | 0.11 (0.02) | 0.22 (0.04) | 0.22 (0.03) |
| 2 | 0.23 (0.04) | 0.23 (0.04) | 0.20 (0.03) | NaN | 0.05 (0.01) | 0.05 (0.01) | 0.22 (0.04) | 0.22 (0.03) |
| 3 | 0.23 (0.04) | 0.23 (0.04) | 0.20 (0.03) | NaN | 0.05 (0.01) | 0.12 (0.02) | 0.22 (0.04) | 0.21 (0.03) |
| 4 | 0.23 (0.04) | 0.23 (0.04) | 0.20 (0.03) | NaN | 0.05 (0.01) | 0.12 (0.02) | 0.22 (0.04) | 0.21 (0.03) |
| 5 | 0.23 (0.04) | 0.23 (0.04) | 0.20 (0.03) | NaN | 0.03 (0.01) | 0.11 (0.02) | 0.22 (0.04) | 0.21 (0.03) |

Table 7: AUROC mean values and confidence intervals for the Heart Attack classification task for random removal followed by non-random removal and feature removal conditions.

|     | MFCL | MFCL+ | Ridge | missForest | GRAPE | GAIN | VAEAC | FC+Mod |
|-----|------|-------|-------|------------|-------|------|-------|--------|
| 0.0 | 0.82(0.05) | 0.73(0.063) | 0.80(0.05) | 0.79(0.05) | 0.83(0.05) | 0.70(0.05) | 0.80(0.05) | 0.83(0.05) |
| 0.2 | 0.81(0.05) | 0.73(0.07) | 0.80(0.05) | 0.77(0.06) | 0.79(0.06) | 0.63(0.06) | 0.81(0.05) | 0.82(0.05) |
| 0.4 | 0.77(0.05) | 0.68(0.06) | 0.76(0.06) | 0.77(0.05) | 0.73(0.07) | 0.49(0.02) | 0.78(0.05) | 0.78(0.05) |
| 0.6 | 0.75(0.06) | 0.66(0.07) | 0.74(0.07) | 0.72(0.06) | 0.68(0.09) | 0.52(0.03) | 0.76(0.06) | 0.78(0.05) |
| 0.8 | 0.69(0.07) | 0.60(0.06) | 0.70(0.07) | 0.64(0.07) | 0.65(0.11) | 0.51(0.04) | 0.72(0.06) | 0.71(0.06) |
| 0.8 | 0.80(0.05) | 0.65(0.06) | 0.78(0.05) | 0.72(0.07) | 0.79(0.05) | 0.50(0.04) | 0.80(0.05) | 0.78(0.05) |
| 0.6 | 0.80(0.05) | 0.66(0.06) | 0.78(0.06) | 0.66(0.08) | 0.79(0.058) | 0.40(0.06) | 0.80(0.05) | 0.78(0.05) |
| 0.4 | 0.79(0.06) | 0.66(0.06) | 0.78(0.06) | 0.65(0.07) | 0.79(0.05) | 0.49(0.04) | 0.79(0.054) | 0.77(0.0) |
| 0.2 | 0.78(0.06) | 0.65(0.06) | 0.77(0.06) | 0.65(0.07) | 0.79(0.05) | 0.48(0.02) | 0.80(0.05) | 0.77(0.04) |
| 1 | 0.82(0.05) | 0.73(0.06) | 0.80(0.05) | NaN | 0.75(0.06) | 0.50(0.0) | 0.81(0.05) | 0.84(0.05) |
| 2 | 0.82(0.04) | 0.74(0.06) | 0.80(0.05) | NaN | 0.63(0.14) | 0.51(0.01) | 0.81(0.05) | 0.83(0.04) |
| 3 | 0.82(0.05) | 0.75(0.06) | 0.80(0.05) | NaN | 0.61(0.15) | 0.50(0.0) | 0.80(0.05) | 0.83(0.05) |
| 4 | 0.81(0.05) | 0.72(0.06) | 0.79(0.06) | NaN | 0.61(0.15) | 0.50(0.01) | 0.79(0.06) | 0.83(0.05) |
| 5 | 0.81(0.05) | 0.74(0.06) | 0.79(0.06) | NaN | 0.64(0.14) | 0.50(0.0) | 0.79(0.06) | 0.83(0.05) |

Table 8: AUPRC mean values and confidence intervals for the Heart Attack classification task for random removal followed by non-random removal and feature removal conditions.

|     | MFCL | MFCL+ | Ridge | missForest | GRAPE | GAIN | VAEAC | FC+Mod |
|-----|------|-------|-------|------------|-------|------|-------|--------|
| 0.0 | 0.05(0.03) | 0.03(0.02) | 0.04(0.02) | 0.04(0.02) | 0.04(0.02) | 0.02(0.01) | 0.04(0.02) | 0.06(0.03) |
| 0.2 | 0.04(0.02) | 0.03(0.02) | 0.04(0.02) | 0.03(0.02) | 0.03(0.02) | 0.01(0.01) | 0.04(0.02) | 0.04(0.02) |
| 0.4 | 0.03(0.02) | 0.02(0.01) | 0.03(0.02) | 0.03(0.02) | 0.02(0.02) | 0.01(0.0) | 0.04(0.02) | 0.04(0.02) |
| 0.6 | 0.03(0.02) | 0.02(0.02) | 0.03(0.02) | 0.03(0.03) | 0.03(0.05) | 0.01(0.00) | 0.04(0.02) | 0.04(0.03) |
| 0.8 | 0.03(0.02) | 0.02(0.01) | 0.03(0.02) | 0.02(0.01) | 0.01(0.01) | 0.01(0.00) | 0.03(0.02) | 0.03(0.03) |
| 0.8 | 0.03(0.02) | 0.02(0.01) | 0.03(0.02) | 0.02(0.01) | 0.02(0.01) | 0.01(0.00) | 0.04(0.02) | 0.03(0.01) |
| 0.6 | 0.03(0.02) | 0.02(0.01) | 0.03(0.01) | 0.02(0.01) | 0.02(0.01) | 0.01(0.00) | 0.04(0.02) | 0.03(0.01) |
| 0.4 | 0.03(0.02) | 0.02(0.01) | 0.03(0.02) | 0.02(0.01) | 0.02(0.01) | 0.01(0.00) | 0.01(0.02) | 0.03(0.01) |
| 0.2 | 0.03(0.02) | 0.02(0.01) | 0.03(0.01) | 0.02(0.01) | 0.02(0.01) | 0.01(0.00) | 0.04(0.02) | 0.03(0.01) |
| 1 | 0.05(0.03) | 0.03(0.02) | 0.04(0.02) | NaN | 0.02(0.01) | 0.01(0.00) | 0.04(0.02) | 0.06(0.03) |
| 2 | 0.05(0.03) | 0.03(0.01) | 0.04(0.03) | NaN | 0.01(0.01) | 0.01(0.01) | 0.04(0.02) | 0.05(0.03) |
| 3 | 0.05(0.03) | 0.03(0.02) | 0.04(0.03) | NaN | 0.01(0.00) | 0.01(0.00) | 0.04(0.02) | 0.05(0.03) |
| 4 | 0.04(0.02) | 0.03(0.02) | 0.04(0.02) | NaN | 0.01(0.00) | 0.01(0.02) | 0.04(0.02) | 0.05(0.03) |
| 5 | 0.04(0.02) | 0.03(0.02) | 0.04(0.03) | NaN | 0.01(0.00) | 0.01(0.00) | 0.04(0.02) | 0.05(0.03) |

Table 9: AUROC mean values and confidence intervals for the Breast Cancer classification task for random removal followed by non-random removal and feature removal conditions.

|  | MFCL | MFCL+ | Ridge | missForest | GRAPE | GAIN | VAEAC | FC+Mod |
|---|---|---|---|---|---|---|---|---|
| 0.0 | 0.91(0.10) | 0.93(0.09) | 0.88(0.14) | 0.52(0.23) | 0.97(0.04) | 0.82(0.15) | 0.81(0.16) | 0.95(0.10) |
| 0.2 | 0.91(0.09) | 0.92(0.09) | 0.87(0.13) | 0.58(0.22) | 0.62(0.16) | 0.69(0.17) | 0.78(0.18) | 0.94(0.09) |
| 0.4 | 0.91(0.10) | 0.89(0.12) | 0.87(0.18) | 0.55(0.26) | 0.51(0.12) | 0.76(0.20) | 0.80(0.20) | 0.90(0.15) |
| 0.6 | 0.89(0.12) | 0.85(0.17) | 0.83(0.15) | 0.43(0.27) | 0.53(0.14) | 0.67(0.22) | 0.75(0.18) | 0.83(0.16) |
| 0.8 | 0.76(0.20) | 0.75(0.19) | 0.68(0.22) | 0.59(0.34) | 0.53(0.09) | 0.73(0.23) | 0.66(0.23) | 0.88(0.13) |
| 0.8 | 0.80(0.10) | 0.77(0.16) | 0.78(0.12) | 0.30(0.17) | 0.94(0.07) | 0.73(0.15) | 0.64(0.13) | 0.83(0.12) |
| 0.6 | 0.80(0.11) | 0.77(0.11) | 0.78(0.11) | 0.40(0.17) | 0.94(0.07) | 0.77(0.13) | 0.64(0.13) | 0.83(0.11) |
| 0.4 | 0.80(0.10) | 0.77(0.11) | 0.78(0.12) | 0.45(0.17) | 0.94(0.07) | 0.70(0.15) | 0.64(0.13) | 0.83(0.11) |
| 0.2 | 0.80(0.11) | 0.77(0.12) | 0.78(0.12) | 0.68(0.17) | 0.94(0.17) | 0.60(0.17) | 0.64(0.13) | 0.83(0.12) |
| 1 | 0.92(0.08) | 0.93(0.08) | 0.90(0.10) | NaN | 0.93(0.08) | 0.79(0.15) | 0.78(0.13) | 0.95(0.07) |
| 2 | 0.92(0.09) | 0.93(0.08) | 0.91(0.10) | NaN | 0.93(0.08) | 0.80(0.14) | 0.78(0.13) | 0.95(0.08) |
| 3 | 0.92(0.08) | 0.93(0.08) | 0.91(0.10) | NaN | 0.92(0.08) | 0.63(0.18) | 0.77(0.13) | 0.95(0.08) |
| 4 | 0.91(0.09) | 0.92(0.08) | 0.88(0.11) | NaN | 0.91(0.09) | 0.72(0.15) | 0.77(0.14) | 0.95(0.07) |
| 5 | 0.91(0.09) | 0.91(0.08) | 0.07(0.08) | NaN | 0.89(0.10) | 0.80(0.14) | 0.77(0.13) | 0.94(0.07) |

Table 10: AUPRC mean values and confidence intervals for the Breast Cancer classification task for random removal followed by non-random removal and feature removal conditions.

|  | MFCL | MFCL+ | Ridge | missForest | GRAPE | GAIN | VAEAC | FC+Mod |
|---|---|---|---|---|---|---|---|---|
| 0.0 | 0.97(0.04) | 0.97(0.04) | 0.95(0.08) | 0.81(0.16) | 0.98(0.03) | 0.94(0.07) | 0.93(0.08) | 0.98(0.05) |
| 0.2 | 0.97(0.03) | 0.97(0.03) | 0.96(0.06) | 0.84(0.14) | 0.70(0.16) | 0.91(0.08) | 0.92(0.09) | 0.98(0.03) |
| 0.4 | 0.98(0.03) | 0.97(0.04) | 0.95(0.09) | 0.82(0.17) | 0.64(0.16) | 0.91(0.12) | 0.93(0.08) | 0.96(0.06) |
| 0.6 | 0.97(0.03) | 0.96(0.06) | 0.96(0.05) | 0.77(0.20) | 0.65(0.16) | 0.92(0.08) | 0.93(0.07) | 0.96(0.05) |
| 0.8 | 0.95(0.07) | 0.95(0.06) | 0.93(0.08) | 0.91(0.13) | 0.64(0.15) | 0.91(0.12) | 0.91(0.11) | 0.97(0.04) |
| 0.8 | 0.86(0.09) | 0.84(0.10) | 0.85(0.09) | 0.55(0.17) | 0.97(0.04) | 0.85(0.11) | 0.74(0.13) | 0.90(0.08) |
| 0.6 | 0.86(0.09) | 0.84(0.10) | 0.85(0.09) | 0.63(0.18) | 0.97(0.04) | 0.87(0.09) | 0.74(0.14) | 0.90(0.07) |
| 0.4 | 0.86(0.09) | 0.84(0.10) | 0.85(0.09) | 0.67(0.18) | 0.97(0.04) | 0.83(0.12) | 0.74(0.13) | 0.90(0.08) |
| 0.2 | 0.86(0.09) | 0.84(0.10) | 0.85(0.10) | 0.77(0.17) | 0.97(0.04) | 0.77(0.14) | 0.74(0.13) | 0.90(0.08) |
| 1 | 0.95(0.05) | 0.96(0.05) | 0.93(0.09) | NaN | 0.95(0.07) | 0.90(0.08) | 0.86(0.10) | 0.97(0.05) |
| 2 | 0.95(0.06) | 0.96(0.05) | 0.94(0.09) | NaN | 0.96(0.05) | 0.90(0.08) | 0.86(0.10) | 0.96(0.06) |
| 3 | 0.95(0.05) | 0.95(0.05) | 0.94(0.08) | NaN | 0.95(0.05) | 0.82(0.13) | 0.86(0.10) | 0.96(0.06) |
| 4 | 0.95(0.06) | 0.95(0.06) | 0.92(0.10) | NaN | 0.94(0.08) | 0.87(0.09) | 0.85(0.11) | 0.96(0.06) |
| 5 | 0.95(0.06) | 0.94(0.06) | 0.95(0.05) | NaN | 0.93(0.09) | 0.90(0.08) | 0.85(0.11) | 0.96(0.05) |

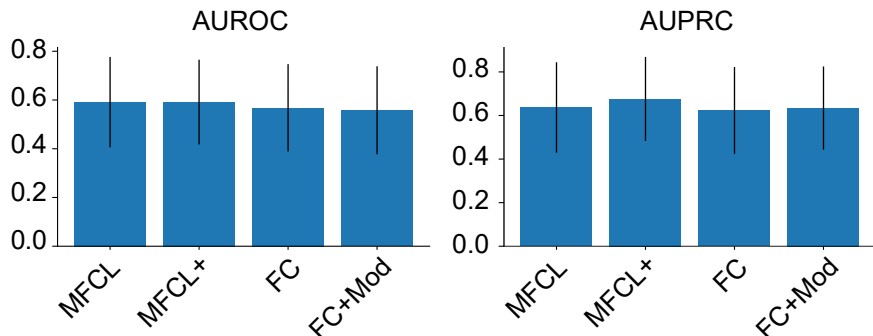

Figure 7: Performance on Breast Cancer classification task with Gaussian errors in predictors (Error bars represent 95% confidence intervals).

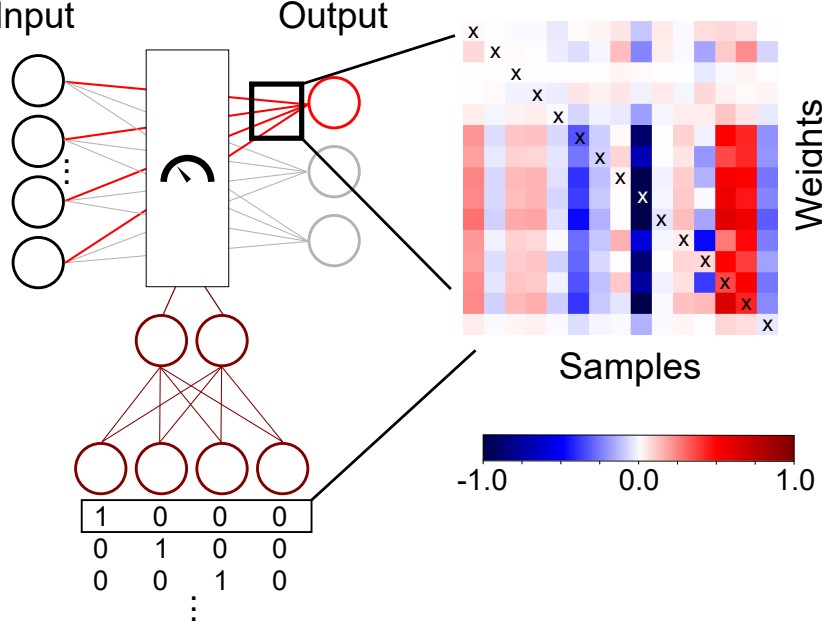

Figure 8: Visualization of modulation of MFCL weights from the Acute Kidney Injury classifier. Here we view the weights highlighted in red. Each column shows a profile of weight change from those when no input is missing where different column shows a different profile based on which input is missing. The x mark denotes the missing value location. Colors show the proportion of change from the original (no-missing input features) weights.

Table 11: Loss mean values and confidence intervals for the Boston Housing Regression task for random removal followed by non-random removal and feature removal conditions.

|  | MFCL | MFCL+ | Ridge | missForest | GRAPE | GAIN | VAEAC | FC+Mod |
|---|---|---|---|---|---|---|---|---|
| 0.0 | 1.02 (0.36) | 1.02 (0.36) | 1.03 (0.35) | 1.02 (0.34) | 1.06 (0.0) | 1.03 (0.34) | 1.02 (0.35) | 1.03 (0.35) |
| 0.2 | 1.02 (0.35) | 1.03 (0.35) | 1.04 (0.34) | 1.04 (0.37) | 1.09 (0.0) | 1.04 (0.31) | 1.04 (0.35) | 1.03 (0.34) |
| 0.4 | 0.95 (0.37) | 0.95 (0.38) | 0.96 (0.37) | 1.01 (0.38) | 1.26 (0.0) | 1.04 (0.37) | 0.96 (0.36) | 0.96 (0.37) |
| 0.6 | 0.97 (0.40) | 0.98 (0.40) | 0.99 (0.38) | 0.96 (0.43) | 1.36 (0.0) | 0.96 (0.42) | 0.99 (0.41) | 0.98 (0.39) |
| 0.8 | 0.92 (0.49) | 0.93 (0.48) | 0.94 (0.47) | 0.98 (0.56) | 1.36 (0.0) | 0.92 (0.48) | 0.94 (0.49) | 0.93 (0.47) |
| 0.8 | 0.96 (0.35) | 0.97 (0.36) | 0.97 (0.34) | 0.98 (0.36) | 1.16 (0.0) | 0.98 (0.34) | 0.97 (0.33) | 0.97 (0.34) |
| 0.6 | 0.96 (0.32) | 0.97 (0.33) | 0.97 (0.31) | 0.98 (0.33) | 1.16 (0.0) | 0.98 (0.31) | 0.98 (0.35) | 0.97 (0.31) |
| 0.4 | 0.96 (0.33) | 0.97 (0.34) | 0.97 (0.32) | 0.98 (0.34) | 1.16 (0.0) | 0.98 (0.33) | 0.97 (0.34) | 0.97 (0.32) |
| 0.2 | 0.96 (0.33) | 0.97 (0.34) | 0.97 (0.33) | 0.98 (0.34) | 1.16 (0.0) | 0.98 (0.33) | 0.98 (0.33) | 0.97 (0.32) |
| 1 | 0.97 (0.33) | 0.97 (0.33) | 0.98 (0.33) | NaN | 1.01 (0.0) | 0.98 (0.33) | 0.98 (0.33) | 0.98 (0.32) |
| 2 | 0.97 (0.33) | 0.97 (0.33) | 0.98 (0.33) | NaN | 1.00 (0.0) | 0.98 (0.33) | 0.97 (0.34) | 0.98 (0.32) |
| 3 | 0.97 (0.32) | 0.97 (0.32) | 0.98 (0.31) | NaN | 0.99 (0.0) | 0.98 (0.31) | 0.98 (0.35) | 0.98 (0.31) |
| 4 | 0.96 (0.35) | 0.97 (0.35) | 0.98 (0.35) | NaN | 0.94 (0.0) | 0.98 (0.35) | 0.98 (0.33) | 0.98 (0.35) |
| 5 | 0.97 (0.35) | 0.98 (0.35) | 0.98 (0.35) | NaN | 1.08 (0.0) | 0.98 (0.34) | 0.98 (0.33) | 0.98 (0.34) |

