# OpenReview forum: "A Modulation Layer to Increase Neural Network Robustness Against Data Quality Issues"
_ICLR.cc/2022/Conference — ICLR 2022 Submitted_

### Official Review · Reviewer_g2de · 2021-10-25

**Correctness:** 3
**Technical Novelty And Significance:** 2
**Empirical Novelty And Significance:** 2
**Recommendation:** 3
**Confidence:** 5

**Main Review:**

*Strengths:

- The text is clear and the motivation is nice. The manuscript provides a very nice introduction to the topic highlighting the importance of the topic.

*Weaknesses:

- The proposed neuromodulation method has three (two theoretical and one practical) main shortcomings: i) it only deals with weights and ignores the biases as a part of network parameters; ii) in its best performance, it is expected to learn masking out the missing values from the inputs that is equivalent to a simple zero-imputation, while adding unnecessary complexity (thus higher instability) into the model architecture; iii) it needs to know the pattern of unreliable inputs at the run time. While finding this pattern is not difficult when the values are missing, it could be a real challenge to find them when the values are of low quality because, for example, the standard deviation of the noise is unknown in many applications (we need an extra module to detect the low-quality inputs in real-time).

- The experimental design can be improved and the results are not decisive and fail in communicating the benefits of the proposed method: The main focus of experiments is on comparing the classification and regression performance between MFCL and its alternatives on a few datasets (only two for classification and one for regression) and the final results show minor or no improvement in many cases. This is while the experimental setup can be improved by a more diverse set of experiments. For example, i) by investigating the patterns learned in $g(m)$; are they simply mask out the missing values or learn something extra? ii) by including more common and simpler imputation approaches such as zero, mean, and KNN imputers. Especially it is very important to show that MFCL is not a simple zero imputer.

*Minor suggestions, comments, and questions:

- The short method section is a little bit inaccurate. For example:
     - "A fully connected layer has a transfer function of": Othe layer types also have transfer function. Furthermore, the formula does not represent a transfer function but the output of a neuron.
     - "and $f$ the non-linearity function": $f$ can be also a linear function. Also 'Non-linearity' should be 'non-linear'.
     - "and fixed at inference": in some models such as Bayesian neural networks, they are not fixed during the inference.
     - the modulation signal $m$ should be in more detail explained in the method section. For example, how to compute it.
     - The definition of $W_0$ is not clear. What is the starting network? Is it the raw network right after initialization?
     - Figure 1 is inaccurate, the modulation network has only two outputs while the number of its outputs should be equal to the number of weights in each layer.

- The proposed method only modulates the weight of the network, thus it still needs to impute the missing values. Throughout the text, it is not clear how missing values are dealt with when MFCL is used. Furthermore, if any imputation is used in advance, then it is difficult to say whether the change in the performance is due to the data imputation of MFCL itself.

- While in the caption, the error bars are missing in Figure 2,3,4.

- The performance of MFCL+ is among the worst in many cases. What is the explanation behind this?

- Are the learned modulation patterns stable across several runs?

- Why other imputation approaches are not benchmarked in the regression task?

- Section 4.2 says the MFCL has a significantly lower loss, but no significance test is performed.

- Sections 4.3 and 4.4 present no quantitative results and miss the proper reference to supplementary materials to guide the reader.


**Summary Of The Paper:**

Inspired by neuromodulation in biological neural networks, this manuscript proposes a modification to fully connected layers in artificial neural networks to accommodate the missing values (or low-quality measures) without the need for suboptimal and unrealistic data imputation. The authors claim that this modification can be useful in applications of AI in real-time settings, but this claim remains unproven (theoretically and empirically) in this paper. The authors benchmarked the proposed methods against a few state-of-art data imputation approaches on two classification datasets and one regression dataset and with random and non-random missing data mechanisms.

**Summary Of The Review:**

Despite good motivation, the proposed method seems to have several theoretical and practical limitations and the current experimental results do not provide enough evidence on what are the benefits of the proposed method.

---

### Official Review · Reviewer_P1JV · 2021-10-30

**Correctness:** 3
**Technical Novelty And Significance:** 2
**Empirical Novelty And Significance:** 2
**Recommendation:** 3
**Confidence:** 3

**Main Review:**

Strengths

1. The paper is well written and easy to read.
2. The approach is well motivated and a good idea.
3. Experiments are thorough and generally well described on a variety of problem types and datasets. I liked the evaluation of confidence intervals.

Weaknesses

1. Unfortunately the method does not significantly improve on the state of the art. In most cases VAEAC does as well or better than the proposed approaches. A simpler version of the proposed approach FC+Mod generally does as well as the more complex MFCL and MFCL+. It was unclear which results showed statistical significance. It seems the networks respond to a fairly simple measure of data quality (FC+Mod), so it is not clear that the more complex approaches are warranted.
2. Many small typographical errors, mainly spelling mistakes. Sometimes FC+Mod is stated as DNN+Mod. Are they the same?
3. Error bars not given on figs 2-4.
4. I am unsure that there is enough information for the results to be reproducible. In particular I was unsure about the network topologies given in the appendix (p13). e.g., for ACTFAST there were 2 hidden layers, with an architecture of 8 neurons in the first layer and 4 in the second, but then I didn't understand how the 3 hidden layers in the next line fitted.
5. It would be good to see a comparison against the approach mostly used of adding additional input attributes for data quality.


**Summary Of The Paper:**

This paper proposes a fully connected neural layer to add to a DNN to deal with missing or noisy data. Weights in the additional neural layer are modulated by quality indicators of the inputs. The neural layer is evaluated in classification, regression and feature imputation settings for  several datasets, one health related one unavailable for replication and the well known Wisconsin Great Cancer and Boston Housing datasets. The approach is evaluated against several state of the art baselines and varying level of data quality. The approach does not do significantly better than existing state of the art methods.

**Summary Of The Review:**

The paper has the good idea of adding a layer modulated on the data quality as a plug and play layer. It is evaluated in a variety of problem types, data sets and experimental settings against state of the art. However, the approach does not do much better that existing state of the art.

---

### Official Review · Reviewer_4EPR · 2021-11-01

**Correctness:** 1
**Technical Novelty And Significance:** 2
**Empirical Novelty And Significance:** 1
**Recommendation:** 3
**Confidence:** 5

**Main Review:**

This submission is framed as for data quality issues in general, but it only tackles missing values.

It is based on an intuition, that modulating the input features based on a function of their reliability, can be helpful. This intuition is interesting. For missing values, NeuMiss networks have showed that similar architectures, with specific choices of modulation and inductive bias, could target the Bayes predictor. However, here, it is given with no analysis and little details about which specific inductive bias to use.

The baseline methods lack strong predictors that readily fit missing values, such as Neumiss or trees with MIA (missing incorporated attribute).

The empirical results are not conclusive.

**Summary Of The Paper:**

This submission contributes an approach to handle missing values in Neural networks by modulating inside the architecture the missing values by factors which decrease  the role of the feature in the architecture. The approach is benchmarked empirically, but does not appear to outperform consistently other approaches.



**Summary Of The Review:**

The contribution is based on intuitions that do not seem very solid and should be better studied. It does not really perform better than other approaches.

---

### Official Review · Reviewer_fZmE · 2021-11-02

**Correctness:** 2
**Technical Novelty And Significance:** 1
**Empirical Novelty And Significance:** 2
**Recommendation:** 3
**Confidence:** 4

**Main Review:**

Empirical evaluation is performed fairly extensively, 3 different kinds of tasks: regression, classification and imputation. Four different - real world datasets from healthcare domain were used, which had natural (label imbalance) as well as simulated data quality issues (noise superposition and masking). The results suggest indeed that introduced "modulation" helped approaches not loose as much performance due to increased degradation of data quality.

In the Related work section, it is stated that "Our approach is superficially similar to attention mechanisms", however not much more effort is made to make a case for that claim. Suggestion that a particular instance of attention mechanism is "difficult to scale for long time-varying inputs" is not a sufficient reason to stop further comparison. Moreover, the scalability argument is brought up, without being accompanied with the analysis which would suggest that proposed "modulation" method is more scalable. The comparison against attention methods, both in terms of computational efficiency/scalability - as well as predictive performance, would have been an a valuable data point.

**Summary Of The Paper:**

The paper is proposing a neural net architecture which makes Weights (in a layer) a function of an external signal trying to quantify the quality of input. The presented architecture is motivated by, and applied to the problem of missing/noisy data. Empirical evaluation performed on different types of tasks, over different datasets and in different missing-ness scenarios showed improved robustness in terms of predictive performance.

**Summary Of The Review:**

Even though empirical study is suggesting potential benefits of the approach in applications with a pronounced data quality issues, the article haven't assured me that proposed modulation layer is more applicable/useful (or sufficiently distinct) from the attention layer. Studies exploring attention for handling missing data have already been conducted (eg. Wu, Richard, et al. "Attention-based learning for missing data imputation in Holoclean." Proceedings of Machine Learning and Systems 2 (2020): 307-325.), and comparison agains them (as a closely related approach) would be appropriate.

---

### Decision · Program_Chairs · 2022-01-20

**Decision:**

Reject

**Comment:**

The paper proposes a modulation layer to address the problem of missing data.

The results do not show that the approach outperforms existing sota approaches.
The results do not demonstrate that the proposed modulation layer is an improvement over attention layer.
Many smaller errors (spelling, etc.) where found in the manuscript.
Experimental details are insufficient to make the results reproducible.
The authors have not provided a response to the reviewers.